# A music structure analysis method based on beat feature and improved residual networks

**Bing Lu**[1], **Qianxue Zhang**[1], **Yi Guo**[1]*, **Fuqiang Hu**[1], **Xuejun Xiong**[2]

**1** Xihua University, Chengdu, China, **2** Mashang Consumer Finance Co, Chongqing, China

* lpngy@vip.163.com

## Abstract

In response to the issues of insufficient audio feature representation and insufficient model generalization ability in music structure analysis methods, a music structure analysis method based on beat feature fusion and an improved residual network was designed. Music structure analysis (MSA) contains two tasks, boundary detection and segment labeling. Boundary detection and Segment labeling are expected to accurately divide music segments and clarify the function. In this paper, a method is studied to accomplish the two tasks. First, the labels of music structure are refactored into 9 types and refined to each beat. Secondly, a beat-wise feature extraction is studied to segment music according to its beats and incorporates various acoustic features to achieve highly accurate segmentation. Then, a Resnet-34 based on a self-attentive mechanism is used to predict the category of each beat. Finally, a post-processing step is designed to filter predicted labels. The method is evaluated on the dataset SALAMI-IA, and experiments show that it is 3 percentage points higher than the current optimal method on HR3F. It is better than the most advanced methods on PWF and Sf.

## 1. Introduction

It is generally believed that music consists of several repetitive segments organized according to a hierarchical structure. Music structure analysis (MSA) is to segment the music into a series of segments and assign these segments to a certain musically meaningful category, such as intro, verse, chorus, interlude, bridge, ending, etc. The tasks of MSA can be divided into boundary detection and segment labeling. The purpose of boundary detection is to segment music into different types of music structures with a certain kind of boundary. Segment labeling aims to assign a category to each music segment obtained by boundary segmentation. MSA can not only be used to deepen the understanding of the music itself, but also can assist many other related fields of research, such as music version recognition, music summary generation, automatic music generation, etc.

A lot of research progress has been made on the boundary detection of music fragments. Paulus et al. [1] distinguished three basic segmentation methods, novelty-based, homogeneity-based, and repetition-based. The novelty-based ones generally become a problem of detecting matrix diagonals after computing self-similarity matrices (SSMs) and self-distance matrices (SDMs). Serrà et al. (2014) [2] used the combination of homogeneity and repetition to

**Data Availability Statement:** All data files are available from the salami-data-public database (https://github.com/DDMAL/salami-data-public).

**Funding:** This work was supported by The 2024 Xihua University Graduate Student Innovation and

Entrepreneurship Competition Project (Grant No.: YK20240013).

**Competing interests:** The authors have declared that no competing interests exist.

generate new structural features to detect boundaries from novelty curves. McFee and Ellis (2014) [3] improved clustering methods using a supervised learning approach for direct learning of feature projections by using the Ordinal Linear Discriminant Analysis (OLDA). Ullrich et al. (2014) [4] used CNNs for supervised learning and found that CNNs are effective in detecting boundaries. Cohen-Hadria (2017) [5] proposed to use a square submatrix centered on the main diagonal of the self-similarity matrix as the input, which was used by a CNN to detect boundaries. Maezawa (2019) [6] used a deep neural network model that incorporates time and three basic methods to improve the accuracy of boundary detection. Hernandez-Olivan (2020) [7] explored the most efficient combination of feature inputs for convolutional neural networks. These segmentation methods have found out the most frequently repeated segments, but the diverse and heterogeneous signals in the same music also led to the poor generalization of the methods.

The various methods mentioned above generally start segment labeling after estimating the boundary. Segment labeling must be done based on boundary detection. It has inspired many scholars to try to combine the boundary detection and segment labeling tasks. Paulus (2009) [8] proposed a complete automatic analysis system using probabilistic fitness measures and greedy search algorithms. Previously, due to the lack of annotated datasets, most studies used unsupervised learning clustering algorithms. McFee (2014, 405–410) [9] also proposed spectral clustering to obtain smaller segments. Shibata et al. (2020) [10] proposed a depth generation method using LSTM for prediction of slices with multiple features after using the traditional hierarchical hidden semi-Markov model (HSMM) [11]. Marmoret et al. (2020) [12] proposed the use of non-negative Tucker decomposition for structural analysis. The unsupervised learning approach is labeling the music with symbols such as "a", "b", "a1", etc., which cannot fully reflect the semantic information of the music. After the proposal of a large annotated dataset, Wang and Lu et al. (2021) [13] proposed supervised metric learning to train a deep neural network (DNN) to extract existing MSA features for segmentation and labeling. Wang and Chen et al. (2021) [14] proposed a supervised multitasking approach to deliberately detect choruses in popular music. Wang and Hung et al. (2022) [15] proposed the first system for direct informs the label prediction of the audio content, which is the most advanced method for chorus detection and boundary detection.

In this paper, in response to the issues of insufficient audio feature representation and insufficient model generalization ability in music structure analysis methods, a complete MSA processing system is designed to accomplish the above two tasks sequentially. Taking into account factors such as the sample size of the dataset used in this paper, the size of the model parameters involved, and hardware conditions, we have decided to use this method for relevant research. Firstly, because existing methods generally use a fixed length of time for segmentation, a feature extraction method divides the time length based on the beat of the music and incorporates various types of time-frequency domain features for analysis. Secondly, a Resnet-34 network based on a self-attentive mechanism is used to predict the category of music segment marked with the beat level. Finally, after completing the classification task for each beat, a smooth filtering post-processing step is used to correct the classification results to further reduce the prediction error.

## 2. Materials and methods

### 2.1 Overall framework

In this paper, a MSA method based on beat feature fusion and improved residual network is proposed, and its system framework is shown in Fig 1. The system framework consists of three main steps: (1) beat feature fusion, (2) segment labeling, and (3) label post-processing.

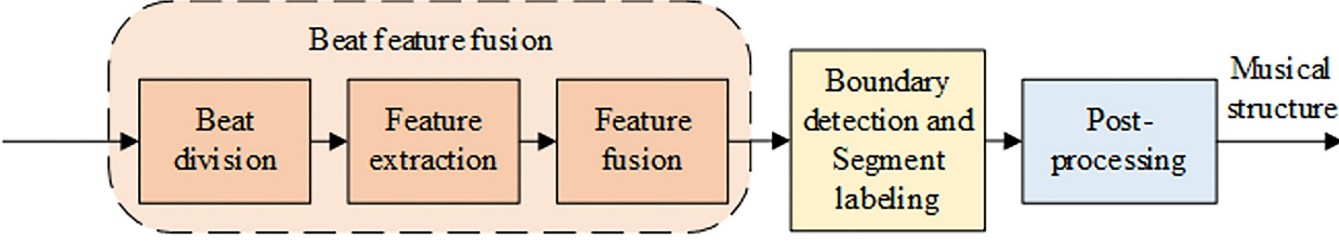

**Fig 1. The system framework of music structure analysis.**

First, in the beat feature fusion step, the music signal is segmented in terms of beat duration, and then 14 acoustic features (such as Constant-Q Transform, Mel-Frequency Cepstral Coefficients, etc.) containing the time-frequency domain are extracted for each beat. Following, a sliding window is used to fusing the features of valid beats. Then, the fused beat features are fed into the improved residual network for segment labeling. Finally, predicted labels are post-processed by label smoothing. Each step is described in detail below.

## 2.2 Data preparation

The data preparation consists of three steps, namely Dataset, label reconstruction, and data pre-processing. These three steps are described in detail next.

**2.2.1 Dataset.** A subset of the Internet of the SALAMI public dataset [16] (Called SALAMI-IA) was used in the experiments of this paper. SALAMI-IA is a database containing a large number of popular, jazz, classical and world music genres. The SALAMI-IA dataset is characterized by its annotated music hierarchy (Lead instrument track, Function track, and music similarity track, respectively). Meanwhile, the size and content of SALAMI-IA dataset is publicly available and downloadable. Therefore, it is used as part of the evaluation dataset by many MIREX music structure segmentation competitions.

The SALAMI-IA dataset is composed of two parts: a CSV file containing the download path and number of the MP3 file about each song, and a folder containing the actual annotations for each song. This paper is specifically focused on the functional structure of the music, so the text file of the functional-level annotations is found in the folder that each song corresponds to. The left column of the text file is the boundary time, and the right column of the text file is the segment label before the boundary time.

Since each song in the SALAMI-IA dataset is annotated by several annotators, the annotation of the first annotator is chosen for each song in order to reduce the ambiguity among different annotators and increase the accuracy of the model.

The annotation of the SALAMI-IA dataset has been continuously revised, and this paper uses the latest corrected version 2.0 completed by the researchers in 2015. In terms of size, the metadata of the SALAMI-IA dataset covers a total of 477 music songs. A total of 448 valid songs were found on the network which is enough to support the training and testing of the model in this paper.

**2.2.2 Labels refactoring.** Functional track in the annotation format of the SALAMI-IA dataset contains 20 music functional labels, which need to be refactored in order to expand the differences between the labels and thus make it easier for the model to distinguish them. As shown in Table 1.

First, a fixed classification defined by Wang [15], in the study of existing datasets, is a seven-category classification suitable for Western popular music. These seven categories are "intro", "verse", "chorus", "bridge", "inst" (i.e., instrumental), "outro " and the auxiliary category

Table 1. SALAMI-IA dataset feature track labeling glossary.

| Group labels | Original Labels |
|---|---|
| Basic group | intro, verse, chorus, bridge |
| Instrumental | instrumental, solo |
| Transition | transition, pre-chorus, pre-verse, interlude |
| Genre-specific | head, main theme, (secondary)theme |
| Form-specific | exposition, development, recapitulation |
| Ending | outro, coda, fadeout |
| Special labels | silence, end |

"silence". These seven categories also apply to the SALAMI-IA dataset, which is equal to separate the four labels within "Basic group", "outro" within "Ending", "silence" within "Special labels" in the SALAMI-IA dataset. At the same time, corresponding to "inst", the labels under the "Instrumental" and "genre-specific" functions are grouped together as "instrumental".

Second, unlike Wang, "interlude" is added in this paper because there is a relatively difference between the label "interlude" and "transition" in the SALAMI-IA dataset, but they are both synonyms of "interlude". Finally, the label "no_function" is added in this paper by playing a few segments with less frequent labels and judging them according to the relationship between their associated sounds. Since not all labels in all parts of a song are functional, and since these labels are between different segment labels, the original labels "applause", "stage_sounds", "spoken", and "crowd_sounds", which appear rarely in the dataset and have low functionality, are added to "no_function".

In this paper, each original label was converted into a total of 9 categories after grouping labels according to the refactoring rule table of labels (Table 2), Separately, they are "silence", "no_function", "interlude ", "intro", "verse", "chorus", "bridge ", "instrumental", "outro."

**2.2.3 Data processing.** After developing the label refactoring rules, in this paper, the Json annotation music specification JAMS format from the Music Structure Analysis Framework (MSAF) [17] is used to formatively analyse each annotation text. Next, the 448 songs in the SALAMI-IA dataset are divided into beats. In this paper, the labels of the SALAMI-IA are refined to each beat according to the label refactoring rules, and the labels of each beat are

Table 2. Specific refactoring rules for the real labels of the SALAMI dataset.

| Refactored label | Original label | Refactored label | Original label |
|---|---|---|---|
| silence | silence | interlude | interlude |
| no_function | no_function | | transition |
| | spoken | | break |
| | stage_sounds | instrumental | instrumental |
| | crowd_sounds | | solo |
| | applause | | theme |
| intro | intro | | variation |
| | head | | main theme |
| verse | verse | chorus | chorus |
| | voice | | pre-chorus |
| | pre-verse | | post-chorus |
| outro | outro | bridge | bridge |
| | fade-out | | build |
| | coda | | |

Table 3. Number of beat samples for each category in the training set, validation set and test set.

|  | Train | Val | Test | Total |
|---|---|---|---|---|
| Silence | 1689 | 450 | 551 | 2690 |
| No_function | 24235 | 6091 | 7609 | 37935 |
| Intro | 14449 | 3721 | 4503 | 22673 |
| Verse | 39245 | 9757 | 12151 | 61153 |
| Interlude | 10295 | 2498 | 3251 | 16044 |
| Instrumental | 55330 | 13828 | 17344 | 86502 |
| Chorus | 31226 | 7776 | 9814 | 48816 |
| Bridge | 4714 | 1200 | 1445 | 7359 |
| Outro | 10253 | 2539 | 3156 | 15948 |

encoded uniquely and thermally. Then, corresponding to each beat is pre-processed according to the method of feature extraction and fusion described previously. At this point, a feature matrix of size 4*424 and a corresponding label are obtained for each beat.

In this paper, the sampling rate of audio signal is 22050Hz, the dot width of FFT windows is 1024, and the size of hop_length is 512 are used to extract the features of each beat. Finally, all beat samples are divided into three parts, which are the training set, validation set and test set. The total number of beat samples in the training set, validation set and test set are 191436, 47860 and 59824, as shown in Table 3.

## 2.3 Beat feature fusion

In this step, the music is segmented based on beat duration firstly, and then feature extraction and fusion is performed for each small segment.

**2.3.1 Beat division.** Firstly, beat division is performed. In MSA, researchers often fix the time of extracting music segments and the length of overlap between segments. Although this can link the context of audio signals, the different characteristics of different types of music signals obviously cannot be satisfied in terms of the representation of detailed features. Meanwhile, the boundaries of musical structure and bars is often the beginning point of a strong beat, and the accurate beat position can also help determine the semantic boundary. Wu et al. [18] and Pachauri [19] have also experimentally demonstrated that classification by beats can improve the accuracy. In this paper, the location of each beat is extracted from the audio using the method introduced by Ellis [20]. The calculation of the number of beats per minute (bpm) and the beats calculation based on dynamic programming are illustrated as follows.

1. Converts input audio to an Onset Strength Envelope

Firstly, the audio file is read at a sample rate of 8kHz. Then, the short-term Fourier transform (STFT) with *window_size* of 32ms and *step_size* of 4ms. Next, the spectrogram is mapped onto the Mel spectrogram with the vertical axis divided into 40 bands. The Mel spectrogram is converted into dB and the first-order difference of each band with time is calculated. The negative values of all differentials are set to 0. The positive values of all differentials at each time point are accumulated.

Finally, the processed signal is passed through a high-pass filter, which filters out information below 0.4 Hz. The onset strength envelope is obtained by smoothing with a Gaussian window with *window_size* of 20ms, which is denoted as O(t).

2. Global tempo estimation (The unit of measure is beats per minute(bpm))

A tempo is a tune in a constant loop, and the period of this loop have to be found out the period of this loop. Specifically, the strength of the beat cycle is as it shown in Eq 1.

$$TPS(\tau) = W(\tau) \sum_t O(t)O(t - \tau) \tag{1}$$

Where, W ($\tau$) is a Gaussian-weighted function on the logarithmic time axis. It is designed to solve the problem of periodic functions with large values of autocorrelation functions on multiples of the fundamental period. The expression is shown in Eq 2.

$$W(\tau) = \exp\left\{-\frac{1}{2}\left(\frac{\log_2 \tau/\tau_0}{\sigma_\tau}\right)^2\right\} \tag{2}$$

Where, $\tau_0$ is the center of the beat period deviation and $\sigma_\tau$ controls the width of the weighting curve (in octaves due to log2). The max of $\tau$ in TPS($\tau$) is the main tempo estimate to be found.

3. Beats based on dynamic Programming

First, C*(t) is defined, which is the state transfer function in dynamic programming, and its expression is Eq 3.

$$C^*(t) = O(t) + \max_{\tau = 0..t-4ms}\left\{\alpha F(t - \tau, \tau_p) + C^*(\tau)\right\} \tag{3}$$

Where, O(t) is the Onset Strength Envelope in step 1, $\alpha$ is the coefficient that balances the two target terms; $\tau_p$ is the period obtained in step 2. F(t-$\tau$, $\tau_p$) is used to measure the difference between the $\tau_p$ and the spacing of the two beats:

$$F(t-\tau, \tau_p) = -\left(\log\frac{t-\tau}{\tau_p}\right)^2 \tag{4}$$

Meanwhile, the path of selected beats can be backtracked by P*(t) to get the final sequence of beats. As in Eq 5.

$$P^*(t) = \arg \max_{\tau = 0, t-4ms}\left\{\alpha F(t - \tau, \tau_p) + C^*(\tau)\right\} \tag{5}$$

In this paper, the duration of each beat is obtained after dividing the song into beats. For example, the song *Hideout\*\* 6:39.mp3* to get its bpm of 123.046875, i.e., the average per beat is 0.4878 seconds. Meanwhile, the song *The Letter.mp3* to get its bpm of 95.703125, i.e., the average per beat is 0.6316 seconds. The split lengths of these two songs were compared with the fixed beat lengths of CNN-Chorus [14] and SpecTNT [15] (in seconds/beat), as shown in Fig 2.

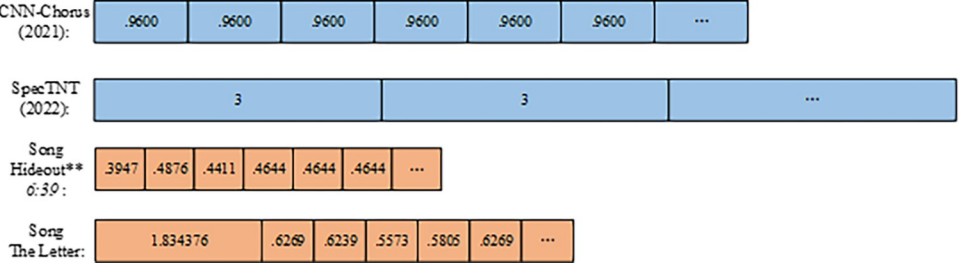

**Fig 2. Minimum sample length comparison chart.**

**2.3.2 Feature extraction.**   After completing the beat division, feature extraction is needed for each beat. The prevailing approaches to MSA all utilize only features extracted based on the music signal itself, and do not capture the musicality characteristics of music, such as the melody and rhythm of music.

Therefore, in order to make a comprehensive consideration for MSA research, 14 time-frequency features that reflect musicality are designed in this paper. The time-domain features include tonal mass features Tonnetz [21] for pitch analysis; the Chroma Energy Normalized Statistics (CENS) that calculates the audio time domain energy[22]; The rhythmic features include Tempogram, which represents the rhythm represented on a time scale [23]. The frequency domain features include the commonly used Constant-Q Transform (CQT) [24], the Mel Frequency Cepstral Coefficient (MFCC) [25], Pitch Class Profile (PCP) [26], The Linear Prediction Coefficient (LPC), Linear Predictive Cepstral Coefficient (LPCC) are also used.

Also, Other spectral features were calculated using the method in  [27], including Gamma Frequency Cepstrum Coefficient (GFCC) reflecting the harmonic structure of the sound signal, Barker Frequency Cepstrum Coefficient (BFCC) reflecting the resonant peak structure of the sound signal, Linear Frequency Cepstrum Coefficient (LFCC) reflecting the fundamental frequency components of the sound signal, Normalized Gamma Chirped Cepstrum Coefficient (NGCC), and Perceptual Linear Prediction (PLP), Rasta PLP (RPLP), which reflect the transient energy of the sound signal and the rate of change of the spectral characteristics.

**2.3.3 Feature fusion.**   After completing the beat division and feature extraction, each beat is fused with features to form a feature fusion, which is then fed into the neural network model, in which a sliding window is used to catch temporal context information. The fusion of beat features refers to combining these beat related features with other acoustic or musical features to obtain a more comprehensive description of the music signal. In music structure analysis, the fusion of beat features is a key process that involves combining different beat features to obtain a more comprehensive and accurate representation of music rhythm. These features may include the intensity, speed, duration, etc. of rhythm, which are important semantic features reflecting different genres and styles of music. The fusion of beat features also plays an important role in music genre classification. By combining beat semantics and MFCC acoustic features, the accuracy of music genre classification can be improved. This indicates that feature fusion not only helps improve the detection and classification of music rhythms, but also enhances the performance of music genre recognition. A sliding window covers four beats, i.e., the feature matrix of beat i is the concatenation of feature vectors of beats i, i-1, i-2 and i +1. This input method enables the model to learn different segments of different songs at the same time. The working of the sliding window during feature fusion is shown in Fig 3.

In this paper, the songs are sequentially passed through the three steps in the beat feature fusion layer. The vector length and matrix size of each feature corresponding to each beat can be obtained. The input of the model is obtained by stitching together all feature matrices of each beat. As seen in Table 4.

## 2.4 Improved Resnet-34

After the feature extraction for each small segment is completed, it needs to be perform boundary detection and fed into a classification network for category labeling. Boundary detection in music structure analysis is a crucial step that involves identifying and locating the starting and ending points of different parts in a musical composition. Boundary detection helps with further analysis tasks such as identifying music structures, classifying music styles, and retrieving music information. Boundary detection is usually achieved by analyzing the characteristics of music signals. These features may include timbre, rhythm, pitch, and other musical attributes.

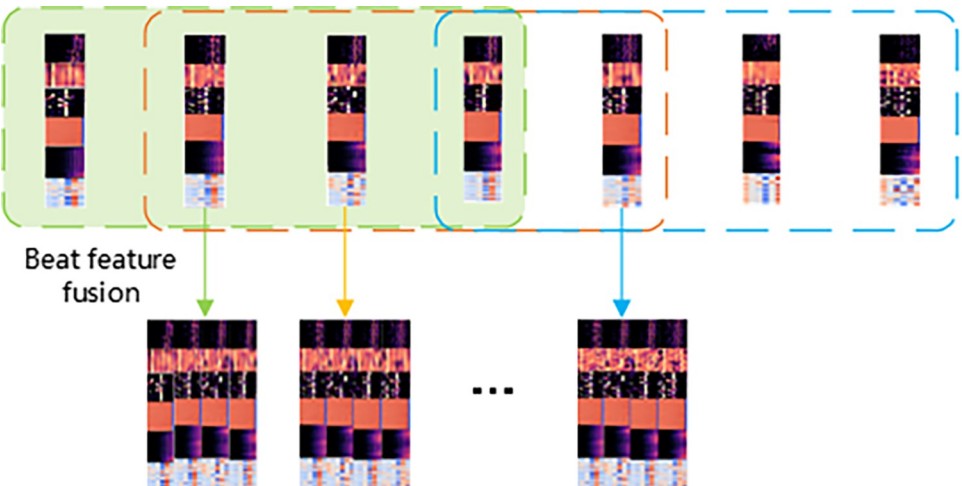

**Fig 3. Feature fusion flow chart.**

The selection of boundary detection algorithms is crucial for the accuracy of the results. Different algorithms may have different recognition effects on music structures, and some algorithms may be more sensitive to boundary detection of certain music styles. In addition, the performance of boundary detection also depends on the quality of the extracted music features and the parameter settings of the algorithm. Some research works attempt to improve the accuracy and robustness of boundary detection by combining multiple features and algorithms. Resnet [28] modified the previous network and added residual units through a short-circuiting mechanism. Residual units are also added to the network structure proposed by Wang [15] (2022) in MSA.

The method studied in this paper uses a baseline model of Resnet-34. In order to focus on the key information in the input features, the self-attentive mechanism module dual attention network module DANet [29] (2019) is added after all layers are completed without changing the residual block structure of Resnet-34. The complete structure of the model in this paper is composed of a convolutional layer and a BN layer, four similar layers, the DANet module, finally the pooling and fully connected layers. As shown in Fig 4.

The layers in ResNet-34 are composed of several consecutive blocks of identical residuals. A residual block directly connected across layers is called a Basic block in ResNet-34. The residual blocks alleviate the gradient disappearance problem brought about while increasing the depth. The structure of Basic block is shown in Fig 5 below.

**Table 4. Vector lengths and matrix sizes for the 14 features.**

| Feature Name | vector length | matrix size |
|---|---|---|
| CQT | 84 | 84×4 |
| CENS | 12 | 12×4 |
| PCP | 12 | 12×4 |
| MFCC | 6 | 6×4 |
| Tonnetz | 4 | 14×4 |
| Tempogram | 192 | 192×4 |
| GFCC,BFCC,LFCC,LPC,<br>LPCC,NGCC,PLP,RPLP | Both 13 | Both 13×4 |
| Total | 424 | 424×4 |

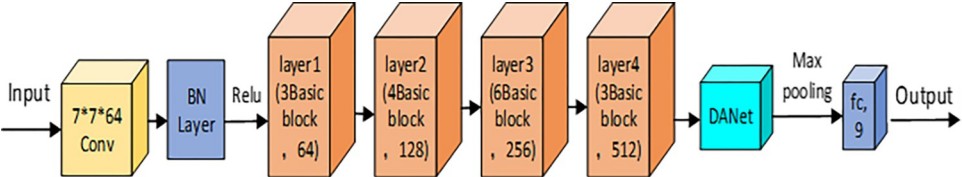

**Fig 4. Residual network structure based on the attention mechanism DANet.**

The classical lightweight DANet was used to improve the baseline model Resnet-34. It allows features to express rich melodic, rhythmic and other information in spatial. It also improves the model's sensitivity to musical details and helps the model to better distinguish the relationship between different segments.

DANet is a fused variant of CBAM [30] and non-local. Compared to SENet [31], which focuses only on channel, CBAM focuses on the channel and spatial dimensions. More importantly, on the idea of CBAM for spatial and channel autocorrelation separately, DANet also directly uses the form of non-local autocorrelation matrices. The position attention module of DANet is concerned with the dependency of any two positions of the feature matrix. It means that similar features will be related to each other regardless of distance. The structure of the DANet module in this paper is as follows Fig 6.

## 2.5 Post-processing of labels

A smooth filtering sliding window to post-process the predicted labels is designed to increase the accuracy of the model prediction. It prevents sudden abnormal jumps in predicted label, as this type of mutation is unlikely to occur in actual music. The sliding window works based on the principle that all predicted labels of beat in a song have continuity. With the exception of the "silence" structure category at the beginning of the song, the structures of the other categories contain at least a few consecutive beats.

Therefore, the length of the label slide window is set to n beats. The follow up experimental results showed that the best filtering effect was achieved when the length was 6 beats, so the method is illustrated here with n = 6. The six beats include the previous beat and the current beat as well as the last four beats.

Starting from the second beat, when the predicted label of beat i agrees with beat i-1, then the predicted label of beat i is the filter label of beat i. When the predicted label of beat i does not agree with the predicted label of beat i-1, the next four beats of the current beat will be scanned. If the prediction label of beat i-1 is detected among the prediction labels of the next four beats, the prediction label of beat i-1 is used as the filter label of beat i. Otherwise, the prediction label of beat i is the filter label of beat i. The whole processing flow is shown in Fig 7.

After the post-processing of labels, the category of each beat can be obtained. The beats of the category change are the boundaries of different structures, so that the boundary detection is done at the same time as the segment labeling.

## 3. Experiments and conclusion

### 3.1 Experimental design

**3.1.1 Experimental environment and parameter settings.** The hardware environment for the experiments is Intel Core i7-11800K CPU and NVIDIA GeForce GTX 3060 GPU. The development language is Python 3.6. The deep learning framework used in this paper is TensorFlow 2.6.0.

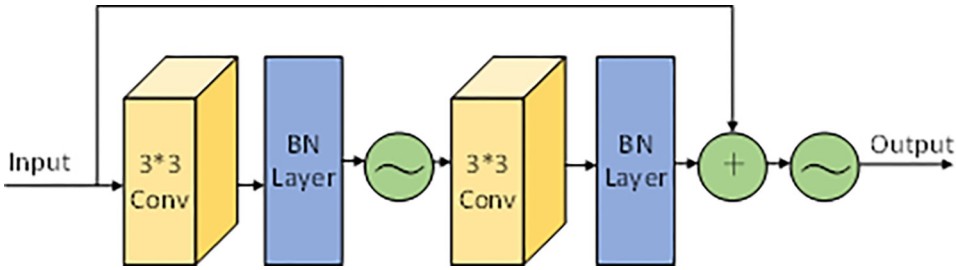

**Fig 5. Structure diagram of the Resnet-34 basic block.**

In this paper, Librosa and spafe are used to extract features. The mir_eval library is used to calculate boundary detection and Segment labeling evaluation metrics in the MSA cross-sectional comparison experiments. Sklearn and matplotlib are used for visualization.

In the initial experiments, a variety of optimizers were used for training. The Adam optimizer with exponential decay function was used uniformly after merit selection, with the initial learning rate adjusted from 0.01 and decaying at a base of 0.96 every 1000 steps. the loss function categorical_crossentropy is used.

**3.1.2 Baseline model selection experiments.** The MSA baseline model selection experiments were first performed. The experimental setup used the same input for each baseline model, i.e., the traditional acoustic features of 14 audio signals extracted manually.

In the field of MSA, convolutional neural networks are most frequently used with great success. So, the baseline models in this paper are mainly based on various classical and improved convolutional neural network (CNN). Meanwhile, because music is temporal in nature, it is essentially a time sequence consisting of musical tones and noise. Thus, some classical recurrent neural networks are also used as the baseline models. For these reasons, the alternative models in this paper include CNN [32]. To be applied in the context of the task of this paper, 14 feature matrices are input to the model. Also included in the baseline model are Alexnet, Resnet, which has shown great strength in music, and Efficientnet [33], which is recently claimed to be the best CNN network. While LSTM can make good use of the information in

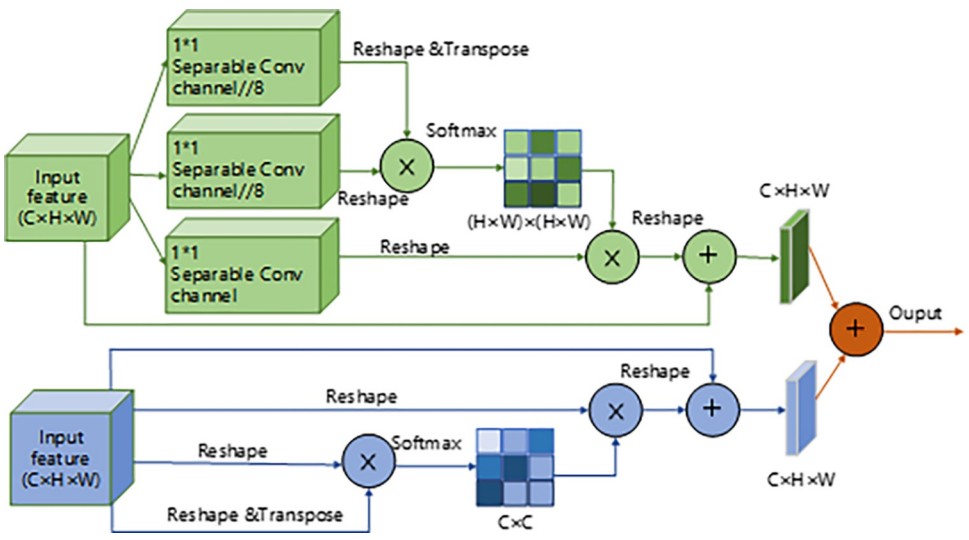

**Fig 6. Structure diagram of the DANet.**

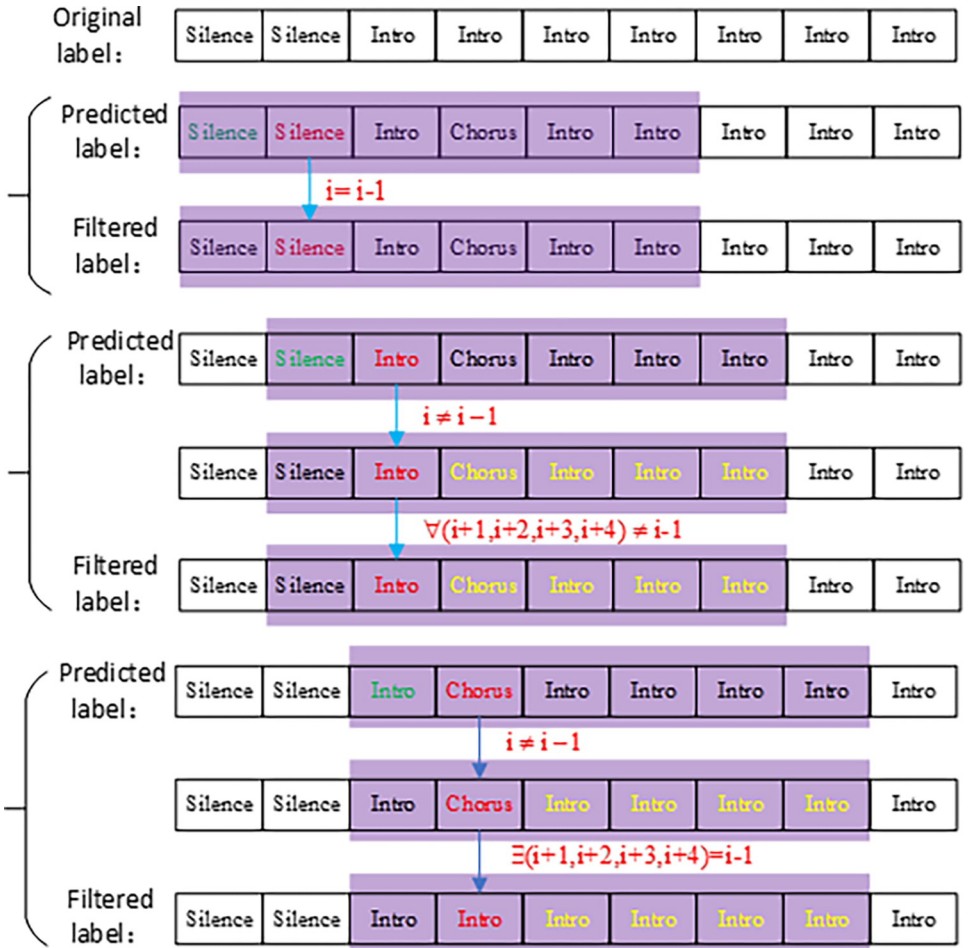

**Fig 7. Complete post-processing diagram.**

the temporal dimension [34], so the improved Multi-BLSTM is also used as the baseline model. The performance of each model is compared by accuracy, recall, and precision, and the size of the model parameters. The baseline model selection experimental setup is shown in Table 5.

**3.1.3 Comparative experiments of attentional mechanisms.** In order to verify the effect of attention mechanisms on the experimental results, two attention mechanisms are added after choosing the baseline model to enable the improved model to achieve better results. The two attention mechanism modules are CBAM and its variant DANet. When combined with Resnet-34, they are added uniformly behind the layer of Resnet-34.

The model inputs are all consistent with section 3.2.2. The three groups of models are (a) Resnet-34, the baseline model, (b) Resnet-34+CBAM, and (c) Resnet-34+DANet. The evaluation metrics are precision, recall, and accuracy. As shown in Table 6.

**3.1.4 Comparative experiments of post-processing effect.** After the above classification task is completed, the improved Resnet-34 model will still have some obvious classification errors when predicting music structures for complete songs. These obvious errors can be eliminated by appropriate post-processing. In a piece of music, the beats belonging to the same structure will not be single, at least several. So, this paper smooths out the abrupt music structure labels by a sliding window. The smoothing effect of sliding windows of different sizes

**Table 5. Different baseline models selection experiment.**

| Input | Baseline Model | Evaluating indicator |
|---|---|---|
| CQT+CENS+PCP+Tonnetz +MFCC+Temprogram +GFCC+BFCC+LFCC+LPC +LPCC+NGCC+PLP+RPLP | CNN | Precision. (%) Recall. (%) Accuracy. (%) Model weight size (MB) |
| | Multi-BLSTM | |
| | Alexnet | |
| | Resnet-18 | |
| | Resnet-34 | |
| | Efficientnet-b0 | |

were compared. The accuracy rate, recall rate and accuracy rate were compared when the sliding window was not used and the length of the sliding window was 6, 7 and 8 beats respectively (i.e., the length of the beat involved in label detection was 4, 5 and 6 beats). As shown in Table 7.

**3.1.5 MSA cross-sectional comparison experiment.** In order to the comparison of the contribution of the models on boundary detection and Segment labeling, some MSA methods are selected for comparison with the methods in this paper. Scluster (a) is a classical approach to clustering. Scluster is capable of handling large-scale datasets due to its relatively low computational complexity. SpecTNT (f) is a multi-point method using spectral-temporal converter + CTL loss-based. SpecTNT demonstrates advanced performance in music tagging and vocal melody extraction. Supervised CNN (b) is a convolutional neural network method for two-stage classification. It plays an important role in multiple fields such as image recognition and object detection. LSTM-HSMM (c) is a hybrid model of hidden semi-Markov model and recurrent neural network. LSTM-HSMM not only offers a profound understanding of sequential data but also facilitates precise prediction and classification of the underlying states within the data.Non-negative Tucker decomposition (NTD) [12] (d) is a non-negative Tucker decomposition. DSF+Scluster [13] (e) is a metric learning method after a supervised approach. This method has significant advantages in fine-grained recognition at the individual level.

After prediction of all SALAMI-IA songs using DANet+Resnet-34 network, the labels are then filtered and post-processed. First, a music specification JAMS file with annotations is created for each song, and then the MSAF framework is used to evaluate the effect on the SALAMI-IA public dataset. As shown in Table 8.

The common assessment metrics for MSA are well defined and detailed information can be found in [35]. In the experiments of this paper, the main focus is on the hit rate metric for boundaries and the clustering metric for Segment labeling. The following evaluation metrics are used in this paper for boundary detection: (1) HR.5F: F-measure of hit rate at a tolerance of 0.5 s; (2) HR3F: F-measure of hit rate at a tolerance of 3 s. The following evaluation metrics are used in this paper for segment labeling: (1) PWF: F-measure of pairwise frame clustering; (2) Sf: F-measure of normalized conditional entropy score. The PWF metric tends to be sensitive to the exact boundary between reference and estimated placement is too sensitive. Sf addresses this problem by employing a probabilistic approach. Sf gives an estimate of the extent to which labels are over or under-segmented.

**Table 6. Comparison experiments of attentional mechanisms.**

| Input | Model | Evaluating indicator |
|---|---|---|
| CQT+CENS+PCP+Tonnetz +MFCC+Temprogram +GFCC+BFCC+LFCC+LPC +LPCC+NGCC+PLP+RPLP | Resnet-34(a) | Precision. (%) Recall. (%) Accuracy. (%) |
| | Resnet-34+CBAM(b) | |
| | Resnet-34+DANet(c) | |

**Table 7. Post-treatment effect comparison experiment.**

| Window size | Model | Evaluating indicator |
|---|---|---|
| NO | | |
| Six beats | DANet +Resnet-34 | Precision. (%) Recall. (%) Accuracy. (%) |
| Seven beats | | |
| Eight beats | | |

## 3.2 Experimental results and analysis

**3.2.1 Baseline model selection.** Different baseline models are selected for the experiments with the same input feature. These baseline models are CNN, Multi-BLSTM, Alexnet, Resnet-18, Resnet-34, and Efficientnet-b0. The accuracy, recall, precision, and model weight size of these models are compared.

As shown in Table 9, the accuracy of CNN performs the worst under the same input. The Multi-BLSTM accuracy under this feature approach ranks second to last Clearly, the Resnet-34 network performs well in all aspects. The deeper network, Efficientnet-b0, did not perform as expected, probably because the context selected in this paper is narrower and the model requires a higher resolution of the feature inputs. Better features and data need to be designed to make the Efficientnet-b0 network more effective. On the dataset used in this paper, the Resnet-34 network has been able to achieve better results with the confusion matrix shown in Fig 8, so it has been chosen as the baseline model.

**3.2.2 Comparison of attentional mechanisms.** Based on the above optimal baseline model Resnet-34 different attentional mechanisms were added for the experiments. Two attention mechanism modules, CBAM and its variant DANet, are used in this paper. The inputs of the three models are the same as in 3.3.1. The experimental results are shown in Table 10.

Comparing the data in the Table 10 shows that the addition of the CBAM model (b) did not produce a significant improvement in the accuracy of the baseline model Resnet-34 (a). In contrast, the addition of the DANet attention mechanism (c) improves the accuracy of Resnet-34 by 1.4 percentage points. The most successful aspect of DANet+Resnet is that the accuracy of the main category "outro" is improved by 4.18 percentage points. The accuracy of "verse", "chorus" and "interlude" is improved by 3.07 percentage points, 2 percentage points and 1.57 percentage points. The results of the attentional mechanism experiment are shown in Fig 9.

**3.2.3 Post-processing effect comparison.** Through the above experiments, the improvement of the accuracy of MSA by the method of beat feature fusion and the introduction of attention mechanism in the baseline model is verified. In order to further avoid prediction errors, in the post-processing experiment, after label prediction is performed on all the beat samples, sliding windows are used to filter the prediction labels on each song. The results of the post-treatment experiments are shown in Table 11.

**Table 8. Comparison results of other MSA models.**

| Model | Dataset | Evaluating indicator |
|---|---|---|
| Scluster (a) | SALAMI | |
| Supervised CNN (b) | SALAMI | |
| LSTM-HSMM (c) | RWC-pop | HR.5F HR3F |
| NTD (d) | RWC-pop | PWF Sf |
| DSF+scluster (e) | SALAMI-pop | |
| SpecTNT (f) | SALAMI-pop | |
| Proposed model | SALAMI-IA | |

**Table 9. Comparison results of different baseline models.**

| Input | Model | Prec. (%) | Rec. (%) | Acc. (%) | Model W_size (MB) |
|---|---|---|---|---|---|
| CQT+CENS+PCP +Tonnetz+MFCC +Temprogram+GFCC +BFCC+LFCC+LPC +LPCC+NGCC +PLP+RPLP | CNN | 82.69 | 75.61 | 82.44 | 44.5 |
| | Multi-BLSTM | 89.16 | 86.62 | 89.17 | 11.1 |
| | Alexnet | 92.84 | 91.88 | 93.41 | 82.5 |
| | Resnet-18 | 92.88 | 92.60 | 93.90 | 2.77 |
| | Resnet-34 | 93.84 | 93.03 | 94.28 | 5.26 |
| | Efficientnet-b0 | 93.21 | 92.43 | 93.86 | 15.9 |

Obviously, the post-processing method proposed in this paper produced significant results. After the beat feature fusion layer and the improved residual network layer, the post-processing with a sliding window length of 6 beats led to an accuracy of 99.14% for the model on the SALAMI-IA dataset in this paper, which is the largest improvement among the three sets of experiments. It can be concluded that it is not the case that the larger the sliding window is, the higher the accuracy is.

The comparison of accuracy without post-processing and with a sliding window of 6 beats for post-processing is shown in Fig 10. Among them, except for the accuracy rate of "no_function" which has largely unchanged, the accuracy rate of the remaining 8 categories has

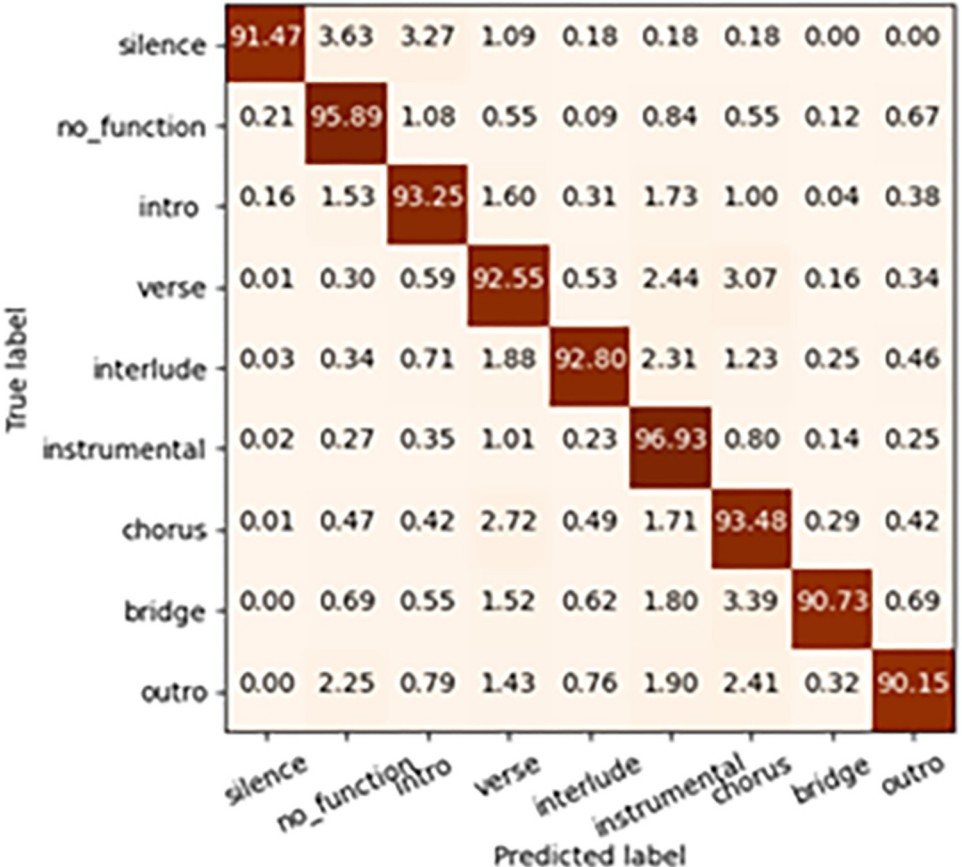

**Fig 8. The confusion matrix plot for the Resnet-34 model.**

**Table 10. Results of the attentional comparison experiment.**

| Model | Prec. (%) | Rec. (%) | Acc. (%) |
|---|---|---|---|
| Resnet-34(a) | 93.84 | 93.03 | 94.28 |
| CBAM+Resnet-34(b) | 93.77 | 93.03 | 94.40 |
| DANet+Resnet-34(c) | 95.08 | 94.30 | 95.68 |

improved significantly. Especially the accuracy rate of the category "silence" has increased by 2.7 percentage points. The accuracy of "outro", "bridge" and "interlude" increased by 1.37, 1.16 and 1.1 percentage points, respectively. Therefore, if the accuracy of the model is to be further improved, follow-up work is needed to distinguish the category "no_function" more carefully.

**3.2.4 MSA cross-sectional comparison.** Finally, a cross-sectional comparison experiment with other works on MSA is conducted. The results of (a), (b) and (d) are directly adopted from [13]. Respectively, the results of (c)and (e) are directly adopted from [10,12,21]. The evaluation results are presented in Table 12.

The model used in this paper is a DANet+Resnet-34 network trained under beat feature fusion. As shown in Table 11, the method used in this paper achieves good results in the evaluation metrics of both the boundary detection and Segment labeling tasks. The outstanding performance of the method on HR.5F and HR3F metrics, which represent boundary detection, demonstrates that the method of refining feature extraction to each beat can be supported by the experimental basis. The HR3F reached to 69.5%, which is 3 percentage points higher than the current best performing method in terms of boundary detection. For example, the bpm of track 1330 is 123.046875, i.e., the length of time between beats is roughly 0.487619 seconds. So, when making predictions for each beat of a song, a hit rate of 3 seconds means that up to 6 consecutive wrong predictions can be made. Although the segment labels are divided into 9 categories in this paper (7 for "SpecTNT"), the accuracy of segment labeling is improved due to the fact that each beat is taken into account. The PWF metric of this model is 0.09 percentage points higher than the best-performing "SpecTNT". The above reason can also explain that the Sf matric exceeds the current top performer "DSF+scluster" by 3.7 percentage points. "DSF+Scluster" also uses the characteristics of musical beats, but does not every beat is refined. This once again validates that the beat-wise approach is effective on segment labeling.

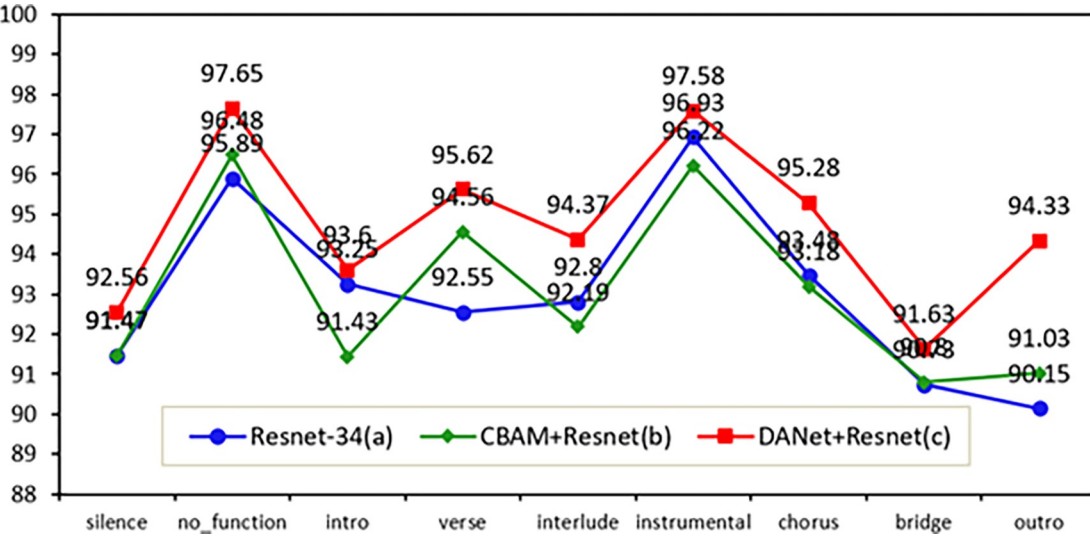

**Fig 9. Comparison of the results before and after adding the attention mechanism.**

**Table 11. Post-processing experimental results.**

| Window size | Model | Prec. (%) | Rec. (%) | Acc. (%) |
|---|---|---|---|---|
| No | | 98.17 | 97.85 | 98.36 |
| Six beats | DANet +Resnet-34 | 98.88 | 99.05 | 99.14 |
| Seven beats | | 98.78 | 98.96 | 99.05 |
| Eight beats | | 98.73 | 98.84 | 98.98 |

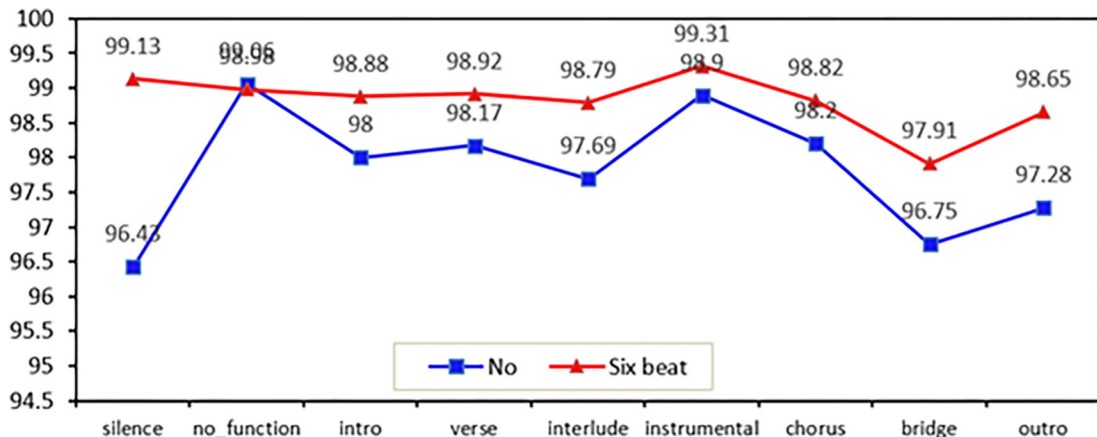

**Fig 10. Line graph comparing accuracy without post-processing and with a sliding window of 6 beats for post-processing.**

**Table 12. Experimental results of cross-sectional comparison between the proposed model and MSA.**

| Model | Dataset | HR.5F | HR3F | PWF | Sf |
|---|---|---|---|---|---|
| Scluster (a) | SALAMI | 0.255 | 0.420 | 0.472 | 0.608 |
| Supervised CNN (b) | SALAMI | 0.541 | 0.623 | 0.505 | 0.650 |
| LSTM-HSMM (c) | RWC-pop | 0.433 | 0.665 | 0.546 | - |
| DSF+scluster (d) | SALAMI-pop | 0.447 | 0.623 | 0.615 | 0.653 |
| SpecTNT (e) | SALAMI-pop | 0.490 | - | 0.651 | 0.632 |
| Proposed model | SALAMI-IA | 0.529 | 0.695 | 0.626 | 0.669 |

# 4. Conclusion and outlook

## 4.1 Conclusion

In this paper, a beat feature fusion processing method is used for music, a MSA method is improved, and a label post-processing method is added to make the automatic MSA effective. Firstly, by using the characteristic of inconsistent beat of different music to segment the music, the musicality features extracted through beat segmentation are more prominent. Adjacent beat features are then fused to form a feature matrix as input in order to link musical contexts. Meanwhile, the DANet attention mechanism is added to the selected baseline model Resnet-34 in order to highlight the correlation between features. Finally, each music is predicted individually and post-processed with labels to smoothly filter out abrupt and erroneous predictions. A cross-sectional comparison study was conducted in this paper, HR.5F is 0.529, HR3F is 0.695, PWF is 0.626 and Sf is 0.669. Most notably in HR3F, which is 3 percentage points higher than the current optimal method, has determined the usefulness of the overall system.

The experimental results show that the end-to-end deep learning model is more effective than feature clustering, and using beats adapted to the song during data pre-processing is more generalizable than a fixed time length cut. The negative is that it is not quite successful in HR.5F, probably because the segments containing boundaries need to be segmented carefully again in this paper.

## 4.2 Outlook

Future work can be carried out in terms of input, model structure and output: (1) Since the rhythms of different music are different, when designing the sliding window size of the feature fusion layer, personalization of music with different rhythms can be tried, and finally the feature dimensions can be standardized. (2) Design different classification structures for different categories of music. (3) Continue to try different classification models. (4) Currently this postprocessing method can only handle well the non-boundary single incorrectly predicted labels, and other methods can be tried to filter the label output in the subsequent work.

## Author Contributions

**Data curation:** Bing Lu, Qianxue Zhang, Fuqiang Hu.

**Formal analysis:** Yi Guo.

**Methodology:** Bing Lu, Qianxue Zhang, Yi Guo.

**Supervision:** Qianxue Zhang, Yi Guo, Xuejun Xiong.

**Writing – original draft:** Bing Lu.

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
