## [Decision Letter · Decision Letter 0]

2 Aug 2024

PONE-D-24-28127A music structure analysis method based on beat feature fusion and improved residual networksPLOS ONE

Dear Dr. Guo,

Thank you for submitting your manuscript to PLOS ONE. After careful consideration, we feel that it has merit but does not fully meet PLOS ONE’s publication criteria as it currently stands. Therefore, we invite you to submit a revised version of the manuscript that addresses the points raised during the review process.

We look forward to receiving your revised manuscript.

Kind regards,

Ali Mohammad Alqudah

Academic Editor

PLOS ONE

Journal Requirements:

   "This work was supported by The Natural Science Foundation of Sichuan Province(No.2023 NSFSC0510, No.2022NSFSC0909 and No.2022NSFSC0490)."

5. We note that your Data Availability Statement is currently as follows: All relevant data are within the manuscript and its Supporting Information files

Reviewers' comments:

Reviewer's Responses to Questions

**Comments to the Author**

1. Is the manuscript technically sound, and do the data support the conclusions?

Reviewer #1: Yes

Reviewer #2: Partly

2. Has the statistical analysis been performed appropriately and rigorously? 

Reviewer #1: Yes

Reviewer #2: Yes

3. Have the authors made all data underlying the findings in their manuscript fully available?

Reviewer #1: Yes

Reviewer #2: Yes

4. Is the manuscript presented in an intelligible fashion and written in standard English?

Reviewer #1: Yes

Reviewer #2: Yes

5. Review Comments to the Author

Reviewer #1: The article "A music structure analysis method based on beat feature fusion and improved residual networks" introduces a new approach to music structure analysis (MSA), focusing on boundary detection and segment labeling. The method refines music structure labels into nine types, segments music based on beats, and extracts various acoustic features for accurate segmentation. It employs a ResNet-34 network with a self-attentive mechanism to predict beat categories and uses post-processing to refine the results. Evaluated on the SALAMI-IA dataset, the method shows a 3% improvement over the current optimal method and outperforms others on PWF and Sf metrics, highlighting the effectiveness of combining boundary detection and segment labeling.

Here are three shortcomings of the article "A music structure analysis method based on beat feature fusion and improved residual networks":

1. Insufficient Comparative Analysis: The article lacks a thorough comparative analysis with more diverse state-of-the-art methods, limiting the context of its contributions and performance.

2. Dataset Limitations: The study only evaluates its method on the SALAMI-IA dataset, which may not fully represent the method's robustness and generalizability across different musical genres and datasets.

3. Complexity of the Model: The proposed method, which includes DANet+ResNet-34 and beat feature fusion, might be too complex for practical applications, and the article does not discuss the computational costs or efficiency of the approach.

References：

E. Jing, H. Zhang, Z. Li, Y. Liu, Z. Ji, and I. Ganchev, "ECG heartbeat classification based on an improved ResNet-18 model," Computational and Mathematical Methods in Medicine, vol. 2021, 2021.

Reviewer #2: Comment: This paper proposes simultaneous boundary detection and segment labeling based on beat

feature fusion. This paper is adequately interesting. However, the crucial problems and proposed solutions

are not well explained in the introduction or abstract. The statement, "The accuracy of segment labeling

will be affected by the accuracy of boundary detection, and the two are inseparable," requires

confirmation. The author explains that existing methods utilize a fixed time length for frame segmentation

in the time-frequency domain, whereas this work segments frames based on beats (introduction: 4th

paragraph, second sentence). However, the subsections on beat division and data processing describe

using fixed window lengths of 32 ms and 1024 samples, respectively. This raises the question of how the

proposed method explains frame segmentation based on beats. The author needs to describe frame

segmentation based on beats more clearly. I have outlined some concerns that require responses as

follows:

1. Please reconsider the title of this paper, as it is too general and not specific. The core objective of

this research is the classification of beat categories.

2. In the introduction, the author argues that the accuracy of segment labeling will affect the

accuracy of boundary detection, and the two cannot be separated (3rd sentence in the 3rd

paragraph). This argument needs to be supported by facts. The previous research described by

the author briefly explains the methods used but does not support the stated argument.

3. The problem and proposed solution should be explained in a structured manner in the

introduction. What is the crucial problem?

4. Abstract, the author does not emphasize the crucial issue of why boundary segmentation and

music labeling based on beats need to be done simultaneously

5. The word “And” cannot be used at the beginning of a sentence.

6. The part explaining boundary detection in Fig.1 required to be visualized.

7. The caption of Fig.1 should be adjusted to the objective of the proposed method.

8. In the last paragraph of the introduction section, 'In this paper, a complete MSA processing system

is designed to accomplish the above two tasks simultaneously.' Be cautious with the term

'simultaneously,' as it implies that the proposed method is end-to-end, meaning it can produce

outputs for both tasks at the same time. However, the framework shown in Fig. 1 is sequential:

The first step performs boundary detection based on beat feature fusion and then uses the output

for segment labeling. Therefore, the term 'simultaneously' is not suitable to describe your

statement."

9. In the abstract and introduction, the author introduces 'boundary detection' as one of the tasks

to be addressed. However, boundary detection is not described in the method section. All terms

mentioned should be clearly described in the method section

10. Preferably section 2 should be Materials and Methods. The dataset can be written in this section.

Is the Data Processing stage related to feature extraction? If yes, combine both subsections.

11. In the titile autor mention ‘beat feature fusion’, however this term is not explained in the method.

12. This paper proposes an improved ResNet-34 by incorporating a lightweight DANet model for beat

category classification. Could you clarify the relevance of ResNet-34 and DANet to your data? For

example, beat detection involves extracting temporal features, therefore, Temporal Convolutional

Network (TCN) is used as proposed model to capture the sequential feature in the music structure.

13. The caption of Figure 4 should be adjusted to match the proposed model name

6. PLOS authors have the option to publish the peer review history of their article (what does this mean?). If published, this will include your full peer review and any attached files.

Reviewer #1: No

Reviewer #2: No

---

## [Author Response · Author response to Decision Letter 0]

19 Aug 2024

Dear editor and reviewers:

First and foremost, we extend our sincere gratitude for the thorough review of our manuscript and for the valuable suggestions and advice provided. We have carefully considered the feedback and have made corresponding revisions to the paper in light of the reviewers' recommendations. Below is our point-by-point response to the reviewers' comments:

The first reviewer's comments:

The article "A music structure analysis method based on beat feature fusion and improved residual networks" introduces a new approach to music structure analysis (MSA), focusing on boundary detection and segment labeling. The method refines music structure labels into nine types, segments music based on beats, and extracts various acoustic features for accurate segmentation. It employs a ResNet-34 network with a self-attentive mechanism to predict beat categories and uses post-processing to refine the results. Evaluated on the SALAMI-IA dataset, the method shows a 3% improvement over the current optimal method and outperforms others on PWF and Sf metrics, highlighting the effectiveness of combining boundary detection and segment labeling.

Here are three shortcomings of the article "A music structure analysis method based on beat feature fusion and improved residual networks":

1. Insufficient Comparative Analysis: The article lacks a thorough comparative analysis with more diverse state-of-the-art methods, limiting the context of its contributions and performance.

2. Dataset Limitations: The study only evaluates its method on the SALAMI-IA dataset, which may not fully represent the method's robustness and generalizability across different musical genres and datasets.

3. Complexity of the Model: The proposed method, which includes DANet+ResNet-34 and beat feature fusion, might be too complex for practical applications, and the article does not discuss the computational costs or efficiency of the approach.

Reply 1：Thank you for the reviewer’s constructive feedback suggesting that the article lacks a thorough comparative analysis with more diverse state-of-the-art methods. We deeply appreciate the reviewer’s vigilance and the opportunity to clarify and correct this issue. We acknowledge that our paper does lack in-depth comparative analysis with advanced methods from different fields. Due to the initial strategy selection in the research, we only selected some music structure analysis methods for horizontal comparative analysis with the methods proposed in this paper in section 3.1.5. To address the issue of limitations on the contribution and performance of the paper due to the lack of comparative analysis, we will provide a detailed explanation of the horizontal comparative experiments in this paper to highlight their context of contributions and performance. The revised content is in paragraph 1 of section 3.1.5.

The additions are as follows:

In order to the comparison of the contribution of the models on boundary detection and Segment labeling, some MSA methods are selected for comparison with the methods in this paper. Scluster (a) is a classical approach to clustering. Scluster is capable of handling large-scale datasets due to its relatively low computational complexity. SpecTNT (f) is a multi-point method using spectral-temporal converter + CTL loss-based. SpecTNT demonstrates advanced performance in music tagging and vocal melody extraction. Supervised CNN (b) is a convolutional neural network method for two-stage classification. It plays an important role in multiple fields such as image recognition and object detection. LSTM-HSMM (c) is a hybrid model of hidden semi-Markov model and recurrent neural network. LSTM-HSMM not only offers a profound understanding of sequential data but also facilitates precise prediction and classification of the underlying states within the data.Non-negative Tucker decomposition (NTD) ‎[12] (d) is a non-negative Tucker decomposition. DSF+Scluster‎[13] (e) is a metric learning method after a supervised approach. This method has significant advantages in fine-grained recognition at the individual level. 

Reply 2：Thank you for the reviewer’s observation regarding the study only evaluates its method on the SALAMI-IA dataset, which may not fully represent the method's robustness and generalizability across different musical genres and datasets.

We acknowledge that our study only evaluates its method on the SALAMI-IA dataset. Due to the incomplete introduction of this dataset in this paper, readers may feel that it may not fully represent the method's robustness and generalizability across different musical genres and datasets. This dataset actually contains different types of music, which is sufficient to support the relevant experiments in this paper. Moreover, this dataset is open source. The revised content is in paragraph 1 of section 2.2.1.

The additions are as follows:

A subset of the Internet of the SALAMI public dataset ‎[30] (Called SALAMI-IA) was used in the experiments of this paper. SALAMI-IA is a database containing a large number of popular, jazz, classical and world music genres. The SALAMI-IA dataset is characterized by its annotated music hierarchy (Lead instrument track, Function track, and music similarity track, respectively). Meanwhile, the size and content of SALAMI-IA dataset is publicly available and downloadable. Therefore, it is used as part of the evaluation dataset by many MIREX music structure segmentation competitions.

Reply 3：In response to the reviewer's question: The proposed method, which includes DANet+ResNet-34 and beat feature fusion, might be too complex for practical applications, and the article does not discuss the computational costs or efficiency of the approach. We provide the following explanation: As shown in Table 10 of the horizontal comparison experiment in section 3.2.4, the DANet+ResNet-34 method proposed in this paper has an average accuracy in analyzing the music structure of various genres in the SALAMI genres dataset. Compared with other methods, it has obvious advantages, which proves the universality and superiority of the music structure analysis method proposed in this paper. This is attributed to the fact that the method proposed in this paper can not only analyze the structure of music from different genres, but also improve the accuracy of music structure analysis. Regarding the lack of discussion on computational cost or efficiency in this paper, we did consider this issue in the early stages of the research. Finally, considering factors such as data sample size, model parameter size, and hardware conditions, we decided to use this method for relevant experiments. We will add relevant discussion explanations to the introduction section. The revised content is in paragraph 4 of introduction.

The additions are as follows:

In this paper, in response to the issues of insufficient audio feature representation and insufficient model generalization ability in music structure analysis methods, a complete MSA processing system is designed to accomplish the above two tasks sequentially. Taking into account factors such as the sample size of the dataset used in this paper, the size of the model parameters involved, and hardware conditions, we have decided to use this method for relevant research.

The second reviewer's comments:

This paper proposes simultaneous boundary detection and segment labeling based on beat feature fusion. This paper is adequately interesting. However, the crucial problems and proposed solutions are not well explained in the introduction or abstract. The statement, "The accuracy of segment labeling will be affected by the accuracy of boundary detection, and the two are inseparable," requires confirmation. The author explains that existing methods utilize a fixed time length for frame segmentation in the time-frequency domain, whereas this work segments frames based on beats (introduction: 4th paragraph, second sentence). However, the subsections on beat division and data processing describe using fixed window lengths of 32 ms and 1024 samples, respectively. This raises the question of how the proposed method explains frame segmentation based on beats. The author needs to describe frame segmentation based on beats more clearly. I have outlined some concerns that require responses as

follows:

1. Please reconsider the title of this paper, as it is too general and not specific. The core objective of this research is the classification of beat categories.

2. In the introduction, the author argues that the accuracy of segment labeling will affect the accuracy of boundary detection, and the two cannot be separated (3rd sentence in the 3rd paragraph). This argument needs to be supported by facts. The previous research described by the author briefly explains the methods used but does not support the stated argument.

3. The problem and proposed solution should be explained in a structured manner in the introduction. What is the crucial problem?

4. Abstract, the author does not emphasize the crucial issue of why boundary segmentation and music labeling based on beats need to be done simultaneously

5. The word “And” cannot be used at the beginning of a sentence.

6. The part explaining boundary detection in Fig.1 required to be visualized.

7. The caption of Fig.1 should be adjusted to the objective of the proposed method.

8. In the last paragraph of the introduction section, 'In this paper, a complete MSA processing system is designed to accomplish the above two tasks simultaneously.' Be cautious with the term 'simultaneously,' as it implies that the proposed method is end-to-end, meaning it can produce outputs for both tasks at the same time. However, the framework shown in Fig. 1 is sequential: The first step performs boundary detection based on beat feature fusion and then uses the output for segment labeling. Therefore, the term 'simultaneously' is not suitable to describe your statement."

9. In the abstract and introduction, the author introduces 'boundary detection' as one of the tasks to be addressed. However, boundary detection is not described in the method section. All terms mentioned should be clearly described in the method section

10. Preferably section 2 should be Materials and Methods. The dataset can be written in this section.Is the Data Processing stage related to feature extraction? If yes, combine both subsections.

11. In the titile autor mention ‘beat feature fusion’, however this term is not explained in the method.

12. This paper proposes an improved ResNet-34 by incorporating a lightweight DANet model for beat category classification. Could you clarify the relevance of ResNet-34 and DANet to your data? For example, beat detection involves extracting temporal features, therefore, Temporal Convolutional Network (TCN) is used as proposed model to capture the sequential feature in the music structure.

13. The caption of Figure 4 should be adjusted to match the proposed model name

Reply 1：Thank you for the reviewer's suggestion that the title of this paper is too general and not specific. The suggestion has important guiding significance for our paper. We have carefully considered your suggestions on the title of the paper and believe that it is necessary to reflect the core content of our research more specifically and accurately. We deeply appreciate the reviewer’s vigilance and the opportunity to clarify and correct this issue. We acknowledge that the original title may have been too broad and failed to clearly convey the main focus of this study. After careful consideration, we have decided to change the title " A music structure analysis method based on beat feature fusion and improved residual networks" to " A music structure analysis method based on beat feature and improved residual networks". In order to more accurately reflect the innovative points and goals of our research. We believe that this new title not only attracts the research interest of peers, but also helps potential readers quickly grasp the main idea of the article. During the revision process, we ensured that other parts of the text were also adjusted accordingly to ensure consistency and coherence throughout the entire article. We expect these improvements to meet the standards of the journal and bring value to the academic community.

Reply 2：Thank you for the reviewer’s insightful question about the argument that the accuracy of segment labeling will affect the accuracy of boundary detection and the two cannot be separated needs to be supported by facts. Reviewer's query has provided us with an opportunity to enhance our exposition on this topic.

We acknowledge that our initial submission did not support the argument through facts. After examining the context of the paper, we have decided to delete this paragraph, which will not affect the structure of the paper.

Reply 3：Thank you to the reviewer for reminding us that the problem and proposed solution should be explained in a structured manner in the introduction. The suggestion that the questions and solutions you raised should be explained in a structured manner in the introduction is very valuable, as it will help improve the logic and clarity of the paper. We acknowledge that our initial submission focused primarily on the conceptual approach and the display of experimental results. Inadvertently overlooking the importance of the raising of the problem in the introduction.Upon careful consideration of your feedback, we have decided to include the missing problem in the introduction to ensure completeness and clarity. We believe that through these adjustments, our introduction will more effectively provide readers with the background, questions, objectives, and expected results of the research, while maintaining the compactness and attractiveness of the introduction.The revised content is in the introduction.

The additions are as follows:

In this paper, in response to the issues of insufficient audio feature representation and insufficient model generalization ability in music structure analysis methods, a complete MSA processing system is designed to accomplish the above two tasks sequentially. Taking into account factors such as the sample size of the dataset used in this paper, the size of the model parameters involved, and hardware conditions, we have decided to use this method for relevant research.

Reply 4：Thank you to the reviewer's careful review and valuable feedback. We deeply appreciate the reviewer’s vigilance and the opportunity to clarify and correct this issue. In response to the reviewer's question: The author does not emphasize the crucial issue of why boundary segmentation and music labeling based on beats need to be done simultaneously in abstract.

In the eighth suggestion given by the reviewer, we understand that these two tasks are not carried out simultaneously, so we have indicated in our response to the eighth suggestion that we will modify this type of description. Music structure analysis (MSA) contains two tasks, boundary detection and segment labeling. Boundary detection and Segment labeling are expected to accurately divide music segments and clarify the function. In this paper, a method is studied to accomplish the two tasks. It can be seen from Figure 1 of this paper. the framework is sequential: The first step performs boundary detection based on beat feature fusion and then uses the output for segment labeling. Finally, a smooth filtering post-processing method was used to correct the classification results, improving the accuracy of music structure analysis. Therefore, we should not emphasize in the abstract that these two tasks need to be carried out simultaneously.

Reply 5：Thank you to the reviewer's careful review and valuable feedback. We sincerely apologize for the grammar error you pointed out about A: The word “And” cannot be used at the beginning of a sentence. Through reading relevant literature and searching for information, we found that 'and' is a rhetorical device used at the beginning to emphasize and emphasize, often used in conversations or speeches. The term 'And' is not suitable for emotionless articles that primarily provide information, as it may appear abrupt in such contexts. We have checked the entire text and

---

## [Decision Letter · Decision Letter 1]

10 Oct 2024

A music structure analysis method based on beat feature and improved residual networks

PONE-D-24-28127R1

Dear Dr. Guo,

We’re pleased to inform you that your manuscript has been judged scientifically suitable for publication and will be formally accepted for publication once it meets all outstanding technical requirements.

Kind regards,

Ali Mohammad Alqudah

Academic Editor

PLOS ONE

Additional Editor Comments (optional):

Reviewers' comments:

Reviewer's Responses to Questions

**Comments to the Author**

1. If the authors have adequately addressed your comments raised in a previous round of review and you feel that this manuscript is now acceptable for publication, you may indicate that here to bypass the “Comments to the Author” section, enter your conflict of interest statement in the “Confidential to Editor” section, and submit your "Accept" recommendation.

Reviewer #1: All comments have been addressed

2. Is the manuscript technically sound, and do the data support the conclusions?

Reviewer #1: Yes

3. Has the statistical analysis been performed appropriately and rigorously? 

Reviewer #1: Yes

4. Have the authors made all data underlying the findings in their manuscript fully available?

Reviewer #1: Yes

5. Is the manuscript presented in an intelligible fashion and written in standard English?

Reviewer #1: Yes

6. Review Comments to the Author

Reviewer #1: authors have adequately addressed comments. A music structure analysis method based on beat feature and improved residual networks

7. PLOS authors have the option to publish the peer review history of their article (what does this mean?). If published, this will include your full peer review and any attached files.

Reviewer #1: **Yes: **zhanlin ji

---

## [Editor Report · Acceptance letter]

14 Oct 2024

PONE-D-24-28127R1 

PLOS ONE

Dear Dr. Guo, 

I'm pleased to inform you that your manuscript has been deemed suitable for publication in PLOS ONE. Congratulations! Your manuscript is now being handed over to our production team.

Kind regards, 

on behalf of

Dr. Ali Mohammad Alqudah 

Academic Editor

PLOS ONE